# How Thymoquinone from *Nigella sativa* Accelerates Wound Healing through Multiple Mechanisms and Targets

Abdalsalam Kmail [1,*], Omar Said [2] and Bashar Saad [3,4,*]

1   Faculty of Sciences, Arab American University, Jenin P.O. Box 240, Palestine
2   Beleaf Pharma, Kfar Kana 16930, Israel; omar@beleafherbal.com
3   Qasemi Research Center, Al-Qasemi Academic College, Baqa Algharbiya 30100, Israel
4   Department of Biochemistry, Faculty of Medicine, Arab American University, Jenin P.O. Box 240, Palestine
*   Correspondence: abdalsalam.kmail@aaup.edu (A.K.); bashar.saad@aaup.edu or bashar@qsm.ac.il (B.S.)

**Abstract:** Wound healing is a multifaceted process necessitating the collaboration of numerous elements to mend damaged tissue. Plant and animal-derived natural compounds have been utilized for wound treatment over the centuries, with many scientific investigations examining these compounds. Those with antioxidant, anti-inflammatory, and antibacterial properties are particularly noteworthy, as they target various wound-healing stages to expedite recovery. Thymoquinone, derived from *Nigella sativa* (*N. sativa*)—a medicinal herb with a long history of use in traditional medicine systems such as Unani, Ayurveda, Chinese, and Greco-Arabic and Islamic medicine—has demonstrated a range of therapeutic properties. Thymoquinone exhibits antimicrobial, anti-inflammatory, and anti-neoplastic activities, positioning it as a potential remedy for skin pathologies. This review examines recent research on how thymoquinone accelerates wound healing and the mechanisms behind its effectiveness. We carried out a comprehensive review of literature and electronic databases, including Google Scholar, PubMed, Science Direct, and MedlinePlus. Our aim was to gather relevant papers published between 2015 and August 2023. The main criteria for inclusion were that the articles had to be peer reviewed, original, written in English, and discuss the wound-healing parameters of thymoquinone in wound repair. Our review focused on the effects of thymoquinone on the cellular and molecular mechanisms involved in wound healing. We also examined the role of cytokines, signal transduction cascades, and clinical trials. We found sufficient evidence to support the effectiveness of thymoquinone in promoting wound healing. However, there is no consensus on the most effective concentrations of these substances. It is therefore essential to determine the optimal treatment doses and the best route of administration. Further research is also needed to investigate potential side effects and the performance of thymoquinone in clinical trials.

**Keywords:** inflammation; wound healing; anti-bacterial; medicinal plants; *Nigella sativa*; natural product

## 1. Introduction

The process of wound healing is complex and involves a sequence of events that work in harmony to maintain equilibrium within the skin and ultimately safeguard the entire body. It involves a complex and dynamic combination of extracellular matrix components, immunological mediators, resident cells, and leukocyte subtypes invading the wound. In the case of skin wounds, the longer it takes a wound to heal, the more opportunities there are for harmful substances/microbial communities to enter the body and cause harm [1]. A wound is a type of injury that results from a variety of causes, including physical, mechanical, chemical, or microbiological factors. These causes can lead to damage to bodily tissues, such as skin epithelial cells, connective tissue, and muscle tissue [2]. Wounds can be classified into several different types, including acute wounds, open wounds, incised wounds, and chronic wounds [3]. The duration of wound healing can be influenced by

various factors such as oxygen levels, presence of infections, age, stress levels, body type, chronic illnesses, vascular insufficiency, and nutritional status. These factors can affect the ability of the body to heal the wound and may result in a delay in sealing and healing the wound [4–7].

The wound-healing process is segmented into four phases: blood clotting and hemostasis, inflammation, cell multiplication, and tissue regeneration (Figure 1) [2]. The first phase of wound healing is hemostasis, which can last up to three days. During this phase, the body initiates blood clotting factors to prevent blood loss. When a wound occurs, the blood vessels in the area of injury constrict, reducing blood flow. Clotting factors are released at the wound site at the same time, triggering fibrin to coagulate and produce a thrombus, or blood clot. This clot forms a barrier between the ruptured blood vessels and helps to prevent further blood loss [8]. The second phase of wound healing is the inflammatory phase, which begins within hours after the wound occurs and can last from day three to day fourteen in the case of acute wounds, with the possibility of lasting longer in the case of chronic wounds. In this stage, inflammation cells like neutrophils and macrophages move to the site of the wound to eliminate dead cells, stop infection, and ready the region for the creation of new tissue. An increase in neutrophils can cause higher inflammation in the wound, which can impede healing. As the inflammatory phase progresses, the number of multinuclear immune cells decreases, while the total number of mononuclear immune cells rises [9,10].

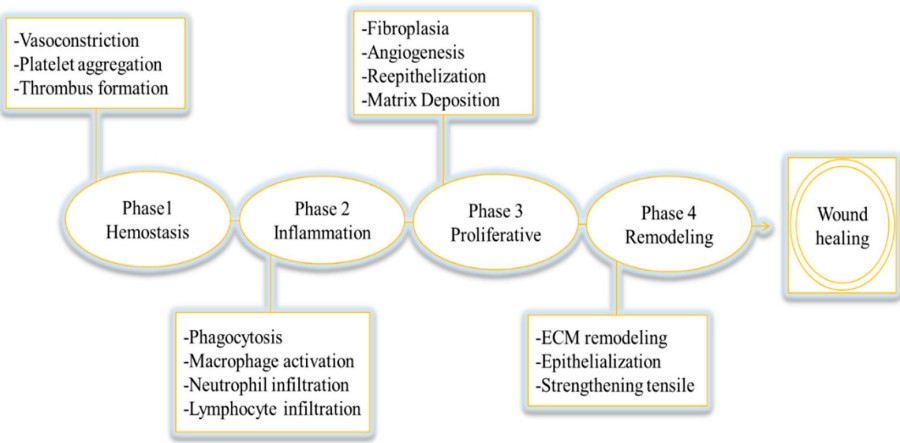

**Figure 1.** Wound healing cascade [8,11].

The third stage of the healing process is the proliferation of connective tissue cells, which lasts from the first to the sixth week after the wound occurs. Granulation tissue, angiogenesis, wound contraction, and epithelialization are all characteristics of this stage. The presence of inflammatory mediators causes the developing tissue to become red or pink. The secretion of collagens by fibroblasts, a kind of connective tissue cell, determines the time it takes for tissue regeneration. During this stage, angiogenesis, collagen accumulation, epithelialization, and the production of granulation tissue all occur to repair skin tissue. Throughout this stage, immune cells decrease at the wound site while fibroblast migration to the wound site increases [10,12]. The final stage of the healing process is marked by tissue scarring and is known as the remodeling or maturation phase. This phase can last from months to years, depending on the severity of the wound, its location, and treatment choices. During this phase, collagen production increases and collagen bundles become more organized. Granulation tissue transforms into scar tissue, other cells are removed through apoptosis, and the wound size continues to shrink. As collagen synthesis continues to enhance the tensile strength and flexibility of the skin, the new tissue gains strength and flexibility [8,10,12].

The challenge of wound healing has been the focus of numerous studies, with many exploring innovative therapeutic strategies and the development of treatment methods for

both acute and chronic wounds in Ayurveda, Chinese, and Greco-Arab herbal medicine. Researchers are exploring new formulas, dressings, and medicinal plant compositions to establish a low-cost, efficient, and sustainable delivery method for wound management/treatment [13,14]. The existence of inflammatory cell types is essential for the production of various cytokines that aid in wound healing, as well as for the gathering and recruitment of fibroblasts to the wound area. Once infectious agents and inflammation have subsided at the end of the first stage of wound healing, vascular regeneration for epithelial tissue construction begins [9]. The effectiveness of a topical medication preparation is greatly influenced by the length of time it is in contact with the skin surface; this requires the preparation to have adhesive properties. Additionally, the effectiveness of topical medications depends on their ability to effectively release the medication [15]. During the proliferation stage of wound healing, fibroblasts migrate and multiply within the wound, where they produce matrix proteins such as hyaluronan, fibronectin, proteoglycans, and collagen fiber. However, research on fibroblasts in diabetic wounds has shown reduced migratory activity and mitogenic responses compared to those from non-diabetic wounds. Additionally, high glucose levels can inhibit the migration and proliferation of keratinocytes, leading to insufficient re-epithelialization [3].

Thymoquinone, the primary bioactive component found in *N. sativa*, boasts a wide range of beneficial properties. Its multifunctional nature has been highlighted in various studies, demonstrating its antimicrobial, anti-inflammatory, anti-allergic, antioxidant, anti-neoplastic, and anti-diabetic capabilities, among others. Notably, thymoquinone has been reported to significantly reduce the tissue damage caused by ischemia-reperfusion. It also exhibits antimicrobial activity and modifies resistance against pathogens. Given these properties, thymoquinone's wound-healing abilities can be attributed to its antimicrobial, resistance-modifying, antioxidant, and anti-inflammatory effects. Furthermore, the therapeutic use of thymoquinone as a wound-healing agent has led to improved fibroblast formation, increased granular tissue production, enhanced wound contraction, and re-epithelization [16–18].

Our review discusses findings from a range of in vitro, animal, and clinical trials that have studied the cellular and molecular mechanisms of *N. sativa*/thymoquinone in wound healing. Therefore, an exhaustive survey of scholarly articles published between 2015 and August 2023 was carried out, utilizing databases such as Google Scholar, PubMed, Science Direct, and MedlinePlus. The aim was to compile data on the healing capabilities of *N. sativa* and thymoquinone in wound recovery. We only considered documents that were published in English in internationally recognized peer-reviewed journals.

We discuss its antioxidant properties, anti-inflammatory effects (including the role of cytokines and signal transduction cascades), antibacterial activity, and its role in accelerating the wound-healing process. The following sections are divided thusly: the use of natural products to regulate inflammation during the healing of wounds; medicinal plants used in the treatment of skin diseases; the traditional uses and active compounds of *N. sativa*; and the beneficial wound-healing effects of thymoquinone. The latter includes the immunomodulatory, antibacterial, and antioxidant effects of *N. sativa* and thymoquinone. Recent advancements in thymoquinone's water solubility and low skin penetration, and the impact of thymoquinone on the balance between wound healing and tissue fibrosis are also discussed. Our review confirms the beneficial impact of *N. sativa*/thymoquinone on wound healing. However, the most effective concentrations of these substances remain a topic of debate. As such, it is vital to establish the ideal dosage for treatment and the most effective method of administration. Furthermore, we need to conduct additional studies to explore any potential side effects and to evaluate the efficacy of these natural products in clinical trials.

## 2. Employing Natural Substances to Control Inflammation in the Process of Wound Recovery

The neutrophils and macrophages recruited phagocytize pathogenic organisms and generate several kinds of highly active antimicrobial compounds such as reactive oxygen species (ROS), cationic peptides, eicosanoids, proteases, and myeloperoxidase (MPO) [19,20]. In addition, neutrophils help to clean up dead tissue by releasing enzymes such as matrix metalloproteinases (MMPs). Monocytes enter the site of injury three days after injury takes place, at which point, they develop into macrophages and contribute to the healing process. The migration of macrophages to the wound location is tightly controlled by gradients of various chemotactic elements, such as growth factors, proinflammatory cytokines, and chemokines [21]. They are important in the wound-healing process. In addition to their immunological functions, they are thought to contribute to a successful outcome of the healing response by synthesizing numerous potent growth factors.

Wounds that take a long time to heal or do not heal properly can become chronic and remain inflamed. These wounds are characterized by a high number of bacteria, abnormal growth factors, inflammatory substances, and enzymes that break down tissue instead of promoting healing. One major consequence of this persistent inflammatory response at the wound site is unbalanced proteolytic activity. Chronic wounds are characterized by an abundance of neutrophils and macrophages, upregulated MMPs, and a prooxidant microenvironment [22].

Medication aids wound healing by influencing various phases of tissue repair. Depending on the mode of action, dosage, and administration method in association with the specific period of wound healing, the medication's effect might be beneficial or harmful. Plant extracts containing natural compounds can offer assistance for enhancing wound healing and minimizing scar formation. Bioactive compounds derived from plants—which have antimicrobial, antioxidant, and healing properties—can stimulate blood coagulation, fight infection, and accelerate wound healing [23–25]. There are many studies that suggest natural substances have the capability to improve wound recovery and can serve as efficient therapies during all phases of the wound-healing process. These compounds can aid in improving the healing process and reducing inflammation and scar formation [26–29]. Phytochemicals are considered safe and are often more tolerated and less expensive than conventional therapies. However, further data from clinical studies on the use of natural products in conjunction with modern medications, as well as the development of improved delivery systems, are required. This is an attractive field for natural product medication advancement [30,31].

Plant extracts and phytochemicals tend to regulate inflammation by acting on wound-healing cells, growth factors, and cytokines. This can result in enhanced angiogenesis, fibroplasia, and epithelialization. In other words, these natural substances can help promote the growth of new blood vessels, the formation of connective tissue, and the regeneration of skin cells, all of which are important for wound healing [24,25]. Although there are challenges in identifying and ensuring the purity of active molecules, there are still a number of clinical studies available on herbal products. These studies provide valuable information on the potential benefits and effectiveness of using plant extracts and phytochemicals in wound healing.

A brief discussion of several promising phytomedicines and their ability to control inflammation during wound healing is presented here. These natural remedies have shown potential in reducing inflammation, reducing microbial infection, and promoting the healing process. For instance, the mucilaginous gel found in the leaves of the *Aloe vera* (*A. vera*) plant has been used since ancient times for its anti-inflammatory and wound-healing properties. The gel's ability to promote wound healing is due to its inhibition of reactive oxygen production, prostaglandins, and cytokines, as well as its ability to enhance the proliferation of fibroblasts and keratinocytes [24,25]. The gel's carbohydrate content activates macrophages and other immune cells involved in the inflammation process. In a rat model of second-degree burns, an *A. vera* solution improved wound healing by

modulating leukocyte adhesion and cytokine levels. Several animal studies have shown that *A. vera* can reduce the severity of acute inflammation. Products such as creams, gels, and impregnated dressings have been tested for their effectiveness in treating acute and chronic wounds in animals [26].

Likewise, honey has been utilized for its wound-healing properties since ancient times. Research has been conducted on various types of honey, with Manuka honey demonstrating significant potential due to its antimicrobial and anti-inflammatory properties [32]. Honey aids in the healing process by creating a moist milieu, removing bacteria and cell debris, providing nutrients and oxygen, and promoting the growth of fibroblasts and endothelial cells. It has been found to efficaciously speed up the healing of both acute and chronic wounds. Honey's anti-inflammatory properties are due to its ability to inhibit factors such as ROS production, the complement pathway, leukocyte migration, COX-2, iNOS, and MMP-9 [33]. Clinical trials have demonstrated that applying honey to wounds can reduce symptoms of inflammation [34].

### 3. Medicinal Plants Used in the Treatment of Skin Diseases

The Middle East is home to a minimum of 2600 types of plants, with 700 of them cited in Greco-Arab medicinal literature for their healing properties. As per multiple studies, more than 450 of these plants continue to be utilized for human disease treatment and are marketed or exchanged both locally and globally. Out of these plants, 40 are presently utilized for treating a variety of skin issues including acne, psoriasis, and allergies [35–37]. Numerous active ingredients have been discovered to be advantageous for the treatment of skin pathologies, as per several in vitro and animal studies. These compounds include β-hydroxychalcone, isoquercetin, ferulic acid, 4-methylthiobutylisothiocyanate, curcumin, sesquiterpene lactone, igalan, linalool, α-peroxyachifolid, quercetin, ursolic acid, saporin, and thymoquinone (Table 1) [13,25].

According to reports, several medicinal plants and their derived active compounds are believed to help hasten the process of wound healing and regenerate tissue at the wound site. Some of the active compounds found in these plants include coelonin, stigmasterol, β sitosterol, anthraquinone derivatives, triterpenoids, rosmarinic acid, crocin, safranal, curcumin, emodin, aloesin, betacyanins, Asiatic acid, thymoquinone, and thymol (Figure 2) [6,24,38–52].

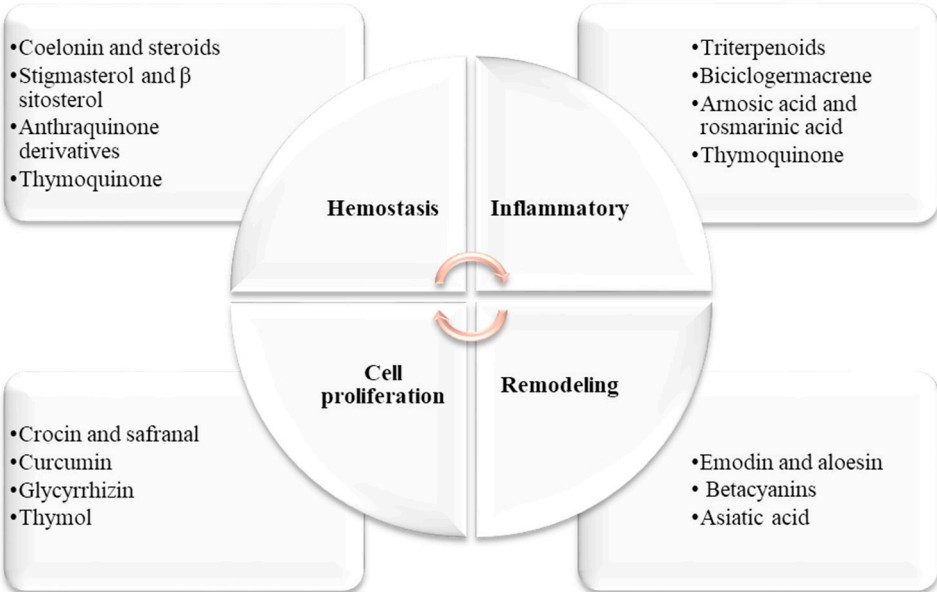

**Figure 2.** Dermatological herbal active compounds and their effects on the four phases of wound healing.

**Table 1.** Active compounds with beneficial effects in skin pathology treatments [6,13,24,25,38–52].

| Plant Name | Common English Name | Parts Used | Active Compound |
|---|---|---|---|
| *Nigella sativa* | Black cumin | Seeds | Thymoquinone |
| *Ficus sycomorus* | Fig-mulberry | Seeds | Isoquercetin |
| *Ferula hermonis* | Lebanese viagra | Rhizome and roots | Ferulic acid |
| *Aloe vera* | Barbadensis Miller | Leaves | Emodin and aloesin |
| *Eruca sativa* | Arugula | Seeds | 4-methylthiobutylisothiocyanate |
| *Curcuma longa* | Turmeric | Rhizome | Curcumin |
| *Inula helenium* | Elecampane | Root | Sesquiterpene Lactone |
| *Linum pubescens* | Perennial flax | Flower | Igalan |
| *Achillea millefolium* | Common yarrow | Flower | Linalool, α-peroxyachifolid, stigmasterol and β-sitosterol |
| *Urtica dioica* | Stinging nettle | Roots | Quercetin and ursolic acid |
| *Saponaria officinalis* | Soapwort | Roots | Saporin |
| *Leptospermum scoparium* | Manuka | Aerial parts | β-hydroxychalcone |
| *Bletilla striata* | Chinese ground orchid | Roots | Coelonin and steroids |
| *Rheum officinale* | Chinese Rhubarb | Roots | Anthraquinone derivatives |
| *Calendula officinalis* | Pot marigold | Flowers | Triterpenoids |
| *Casearia sylvestris* | Crackopen | Leaves | Biciclogermacrene |
| *Rosmarinus officinalis* | Rosemary | Essential oil | Arnosic acid and rosmarinic acid |
| *Crocus sativus* | Saffron crocus | Aerial parts | Crocin and safranal |
| *Glycyrrhiza glabra* | Licorice | Roots and leaves | Glycyrrhizin |
| *Hylocereus undatus* | Dragon fruit | Fruits | Betacyanins |
| *Centella asiatica* | Asiatic pennywort | Aerial parts | Asiatic acid |

### 3.1. Traditional Uses and Active Compounds of N. sativa

*N. sativa*, also known as black cumin or black seed, is an herbaceous plant belonging to the Ranunculaceae family. It produces an annual flower with five to 10 petals that can be white, yellow, pink, light blue, or lavender in color. The plant's fruits are large swollen capsules that contain three-dimensional white seeds. These seeds turn black after being exposed to air and have a fragrant and bitter flavor [53–55]. For many centuries, *N. sativa* has played a significant role in the culture, cuisine, and traditional medicine of South Asia, as well as in Europe and the Mediterranean region (Figure 3) [56].

*N. sativa* is widely known and used in the Arab world, where it is called kalonji and Habbatul Barakah, meaning "seed of blessing" in Arabic. The seeds are often used as an ingredient in bread and pastries, and a dessert made from it is prepared by mixing it with flour and roasting it in the oven (Figure 3) [13,57]. Historical records from various scientific and religious texts have documented the curative properties of *N. sativa*. According to Islamic Hadith, the Prophet Muhammad instructed others to "use the black seed, for without a doubt, it is a cure for all sicknesses except death." Avicenna (980-1037 AD), an ancient Muslim physician and philosopher, mentioned *N. sativa* in his famous "Canon of Medicine", which was part of the traditional medical curriculum from the 12th to the 17th centuries [58,59]. In his canon of medicine, Avicenna described the seeds as "the seed that enhances the energy production in the body and facilitates recovery from fatigue and dejection", weariness, fever, headache, skin ailments, wounds, fungus, parasites, and deadly animal attacks [59–63].

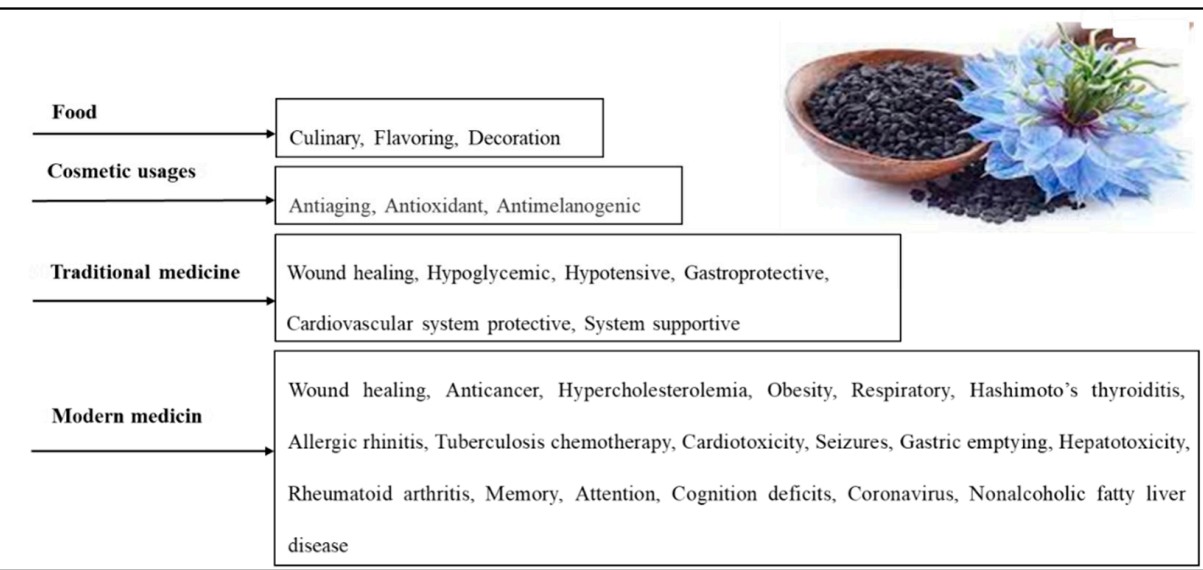

**Figure 3.** Various *N. sativa* seed applications [57].

The principal constituents of *N. sativa* seeds are fixed oil—33.3%, proteins—23.45%, carbohydrates—31.8%, crude fiber—7.2%, minerals—3.3%, and essential oil—0.95%. The therapeutic effects of the seeds are attributed to the main active compounds thymoquinone, dithymquinone, thymohydroquinone, and thymol. Other chemicals found in the seeds comprise nutritious components including carbohydrates, lipids, vitamins, minerals, and vital amino acids. They contain a high concentration of essential and unsaturated fatty acids such as linoleic acid and oleic acid. Phosphatidylcholine, phosphatidylethanolamine, phosphatidylserine, and phosphatitdylinisitol are also found in the seeds. The seeds also include calcium, iron, potassium, and carotene, which the liver converts into vitamin A [64,65].

The surge in interest in phytomedicine has raised concerns about their safety and the legal obligations to comply with health standards [66]. *N. sativa* is "generally recognized as safe" by the FDA in the United States and is available in various therapeutic formulations such as dietary supplements, oils, topical creams, and powders [56]. Numerous clinical trials and research have been conducted to examine its effects on different medical conditions, with evidence indicating tolerable and effective dosages of roughly 1–3 g of powder once day, 40 mg/kg of oil once daily, or 1 mL of topical cream three times per day [67,68]. A topical administration of *N. sativa* cream, oil, or hydrogel is optimal for cutaneous dermatological problems, whereas oral oil or supplement formulations are suitable for systemic dermatological disorders. Although no significant negative effects of the oral oil supplements, topical cream, or hydrogel formulations have been observed, three women who used the topical oil formulation had acute contact dermatitis [56,67,68]. A recent phase 1 trial—which was randomized, double blinded, and placebo controlled—assessed the safety of a *N. sativa* oil formulation containing 5% TQ (BCO-5). The trial administered a dose of 200 mg/adult/day for 90 days to healthy subjects (n = 70). The study analyzed both biochemical and hematological parameters, along with any adverse events or side effects, to determine the clinical safety of BCO-5. The study found no serious adverse side effects or significant changes in the hematological parameters. Similarly, no significant changes were observed in the liver function (ALT, AST, ALP) and renal function (serum creatinine and urea). However, a significant reduction in total cholesterol, LDL, VLDL, and triglycerides was noted in the lipid profile analysis, although these values remained within the normal range. Therefore, BCO-5 should be clinically evaluated for various health beneficial pharmacological activities at a dosage of 200 mg/adult/day [69,70].

The dermatological benefits of *N. sativa* and thymoquinone are attributed to their potent antioxidant, anti-inflammatory, antibacterial, and immunomodulatory properties, making them suitable choices for the treatment of several skin diseases (Figure 3). Nasiri

et al. [71] conducted a meta-analysis of 14 randomized controlled trials to evaluate the effectiveness of *N. sativa* products for treating various skin conditions. The participants of these trials had different types of skin diseases, such as psoriasis, eczema, and acne. The average age of the participants was 28.86 (4.49), and 69% of them were female. The duration of the treatment ranged from 4 days to 24 weeks. A meta-analysis showed that *N. sativa* products had a significant effect on improving the symptoms of skin diseases, with an odds ratio of 4.59. Most of the trials used *N. sativa* essential oil or extract, applied either orally or topically. The authors concluded that *N. sativa* products could be used as an alternative therapy for skin problems, but more research is needed to confirm their efficacy and safety.

*3.2. The Beneficial Wound-Healing Effects of Thymoquinone*

*N. sativa* and thymoquinone have been shown to have a wide range of therapeutic benefits (Figure 4). These characteristics, which include anti-inflammatory, anti-allergic, antioxidant, anticancer, and antibacterial properties among others, have been proven through a variety of in vitro and animal studies [8,57,72]. The positive effects of thymoquinone on skin wound healing are primarily due to the induction of angiogenesis, anti-inflammatory, antioxidant, and antibacterial effects, as well as an increase in fibroblast proliferation and collagen synthesis. These processes work together to promote the healing of skin wounds (Figures 4 and 5) [24,25].

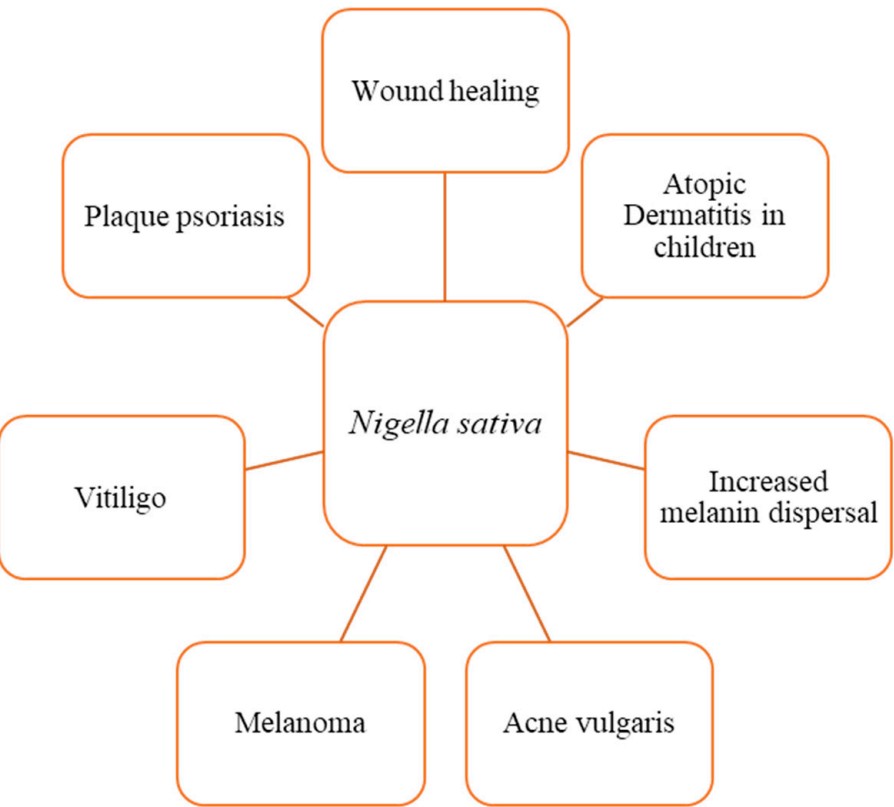

**Figure 4.** Dermatological effects of *N. sativa* and its active compound, thymoquinone. Several in vitro and animal studies have recognized *N. sativa* and thymoquinone for their extensive therapeutic beneficial effects, which stem from their anti-inflammatory, anti-allergic, anti-diabetic, antioxidant, anticancer, antibacterial, nephroprotective, and neuroprotective characteristics.

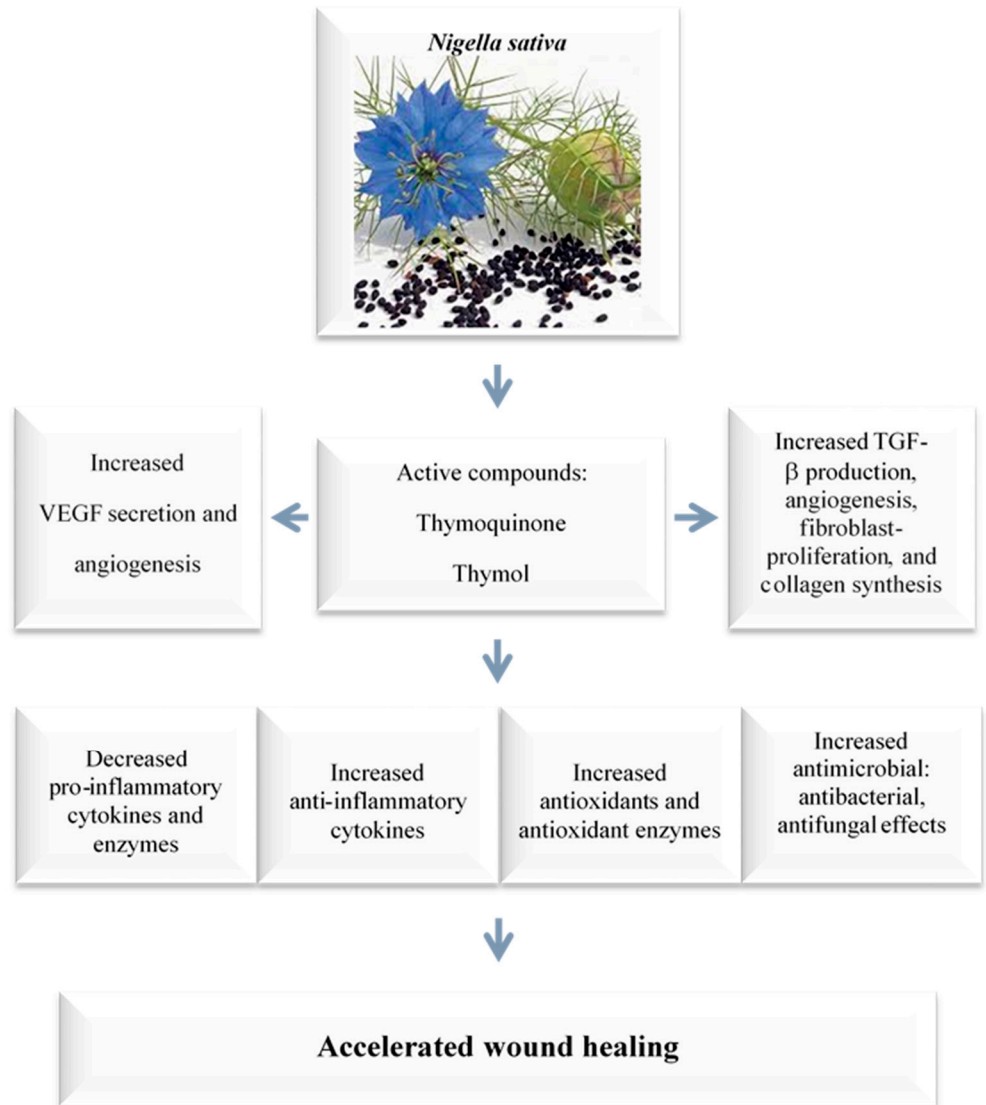

**Figure 5.** Wound-healing mechanisms of *N. sativa* and thymoquinone. The positive effects of thymoquinone on skin wound healing primarily stem from its antibacterial effects; the induction of angiogenesis, anti-inflammatory, and antioxidant properties; as well as an increase in fibroblast proliferation and collagen synthesis. These processes collectively promote the healing of skin wounds. Vascular endothelial growth factor (VEGF); Transforming growth factor beta (TGF-β).

Thymoquinone has been shown to significantly reduce the tissue damage caused by ischemia-reperfusion, a condition in which blood flow is temporarily restricted and then restored to an organ or tissue [73]. In addition, thymoquinone has also been shown to have antibacterial and resistance-modifying effects against a variety of infections [4,5]. It has also been shown to boost anti-inflammatory effects, reduce oxidative stress, enhance fibroblast proliferation, increase granular tissue growth and development, speed up wound contraction, and improve re-epithelization [44,74–76]. Selcuk et al. [74] investigated the wound-healing activity of thymoquinone in a rat model and concluded that thymoquinone's wound-healing activities are due to its anti-microbial, resistance-modifying, antioxidant, and anti-inflammatory effects [44,72–76]. Applying thymoquinone topically to wounds is an effective approach because it allows for direct penetration to the affected areas [77]. However, the therapeutic effectiveness of thymoquinone is limited since it has low water solubility and low skin penetrability, as well as restricted systemic availability. Despite these limitations, thymoquinone has shown promise for use in medical applications due to its multifunctional activity [78,79].

In a study examining the safety of thymoquinone for patients with advanced refractory malignant disease, it was found that thymoquinone was well tolerated in doses ranging from 75 mg/day to 2600 mg/day. The study did not report any toxicities or therapeutic responses [80].

### 3.2.1. Thymoquinone's Effects on Inflammation and Immune System Modulation

Immunomodulation is the alteration of the body's innate and adaptive immune responses, which involves managing the communication between different components like neutrophils, macrophages, and T and B-cells. Immunomodulators, which can either stimulate or suppress the immune system through cellular interactions and paracrine and autocrine mechanisms, play a crucial role in supporting immune function [81].

The innate immune system is made up of numerous protective strategies that act as the initial barrier against potential threats. Cells within this system produce pro-inflammatory cytokines, which boost the body's defense mechanisms by generating antimicrobial mediators, the migration of leukocytes, and creating a protective environment that shields tissues from microbial invasions. During the later stages of the inflammatory phase in the wound-healing process, it has been observed that macrophages transition from a pro-inflammatory to an anti-inflammatory phenotype, a topic we will further delve into. The innate immune system is capable of rapidly activating and launching responses to harmful agents and their products, which can range from pathogen-associated molecular patterns (PAMPs) to damage-associated molecular patterns (DAMPs). The immune system's pattern recognition receptors, including Toll-like receptors (TLR), identify these unique patterns, thereby triggering cellular defenses. These defenses include pro-inflammatory cascades against internal or external danger signals, foreign entities such as viruses and bacteria, and particles. For a more detailed review of innate immunity in chronic wounds, please refer to the following section [82,83].

In essence, inflammation is a crucial, nonspecific response of the innate immune system. This innate immune system not only interacts with but also guides and instructs the adaptive immune system to produce the most effective immune responses [84]. In addition, the reaction of the innate immune system is closely linked to the production of immune-mediators such as interferons (IFNs), interleukins, and antimicrobial peptides and proteins (AMPs), along with acute-phase proteins. The innate immune system comprises macrophages, neutrophils, mast cells, eosinophils, and innate lymphoid cells. The subsequent subsections provide a more detailed explanation of these cells' role in wound healing. Figure 3 illustrates the different immune cells involved and their respective roles and interconnections in the wound-healing process [84].

The immunomodulatory properties of thymoquinone have been a significant area of research for many decades, with the goal of enhancing the human immune system [85,86]. Investigations into the immunomodulatory effects of thymoquinone have been conducted in both animal studies and under in vitro conditions. These studies have shown that thymoquinone has the ability to control the development and cellular reactions of various immune cells, such as T-cells, B-cells, macrophages, neutrophils, NK cells, and dendritic cells [87]. Research on thymoquinone has indicated its potential effects on gestational diabetic females. It has been suggested that thymoquinone can reverse the reduced production of interleukin-2 (IL-2) and enhance the production of T-cells, thereby maintaining T-cell-mediated immune responses [85]. In another study, it was found that a small amount of thymoquinone can enhance the survival of T-cells during CD85 activation and CD62L expression. This led to the conclusion that thymoquinone could be a potent agent in combating infectious diseases. Additionally, it was noted that thymoquinone could enhance T-cell activation, thereby improving adaptive immunity [85]. Moreover, there is compelling evidence that thymoquinone has a modulatory effect on the nuclear factor erythroid 2-related factor 2 (Nrf2). This effect is reported to occur through the inhibition of NF-κB signaling pathways [86].

The role of thymoquinone in innate immunity is well defined, as it has been demonstrated to inhibit the maturation of dendritic cells induced by lipopolysaccharides. This is achieved through the activation of the caspase cycle, along with various interleukins and TNF-$\alpha$ [86]. Moreover, thymoquinone has the ability to halt and reverse the reduction in leukocyte count and immunoglobulin levels caused by pesticides, as it enhances phagocytic activation by stimulating macrophages. These studies highlight the inherent characteristics of thymoquinone as an immunomodulatory agent, capable of effectively stimulating innate immunity through the activation of various immune cells. While these reports do not directly link thymoquinone to anticancer immunity, they do successfully underscore the immunomodulatory properties of thymoquinone and its potential as an immunotherapy agent [85–89].

With the aim of elucidating the mechanisms of the anti-inflammatory and antioxidative activities of thymoquinone, Kundu et al. [90] conducted research to clarify the anti-inflammatory and antioxidative effects of thymoquinone at the molecular level. They discovered that when hairless mouse skin was pretreated with thymoquinone, it reduced the expression of COX-2 induced by 12-O-tetradecanoylphorbol-13-acetate (TPA). Moreover, it was discovered that thymoquinone can lessen the nuclear translocation and DNA binding of NF-$\kappa$B by inhibiting the phosphorylation and subsequent degradation of I$\kappa$B$\alpha$ in TPA-treated mouse skin. Thymoquinone pretreatment also reduced the phosphorylation of Akt, c-Jun-N-terminal kinase, and p38 mitogen-activated protein kinase but not extracellular signal-regulated kinase-1/2. The topical application of thymoquinone led to an increase in the expression of heme oxygenase-1, NAD(P)H-quinoneoxidoreductase-1, glutathione-S-transferase, and glutamate cysteine ligase in mouse skin. The authors inferred that thymoquinone's capacity to suppress TPA-induced COX-2 expression and NF-$\kappa$B activation, along with its ability to stimulate the expression of cytoprotective proteins, forms the mechanistic foundation for its anti-inflammatory and antioxidative effects (Figure 5).

In recent research conducted by Hijazy et al. [91], the effects of oral administration on the development of edema, oxidative stress, and inflammation in mice that had paw edema were investigated. The study found that thymoquinone reduced paw edema volume over time, lessened writhing movements caused by acetic acid, and decreased ear edema triggered by xylene. Hematological results showed the significant normalization of altered WBC and platelet counts. Additionally, paw tissue levels of malondialdehyde and nitric oxide decreased significantly, while Nrf2, glutathione, superoxide dismutase, catalase, glutathione peroxidase, and glutathione reductase increased after thymoquinone administration. Thymoquinone has been shown to reduce pro-inflammatory mediators in inflamed paw tissue, including IL-1, TNF-$\alpha$, IL-6, MCP-1, C-reactive protein, myeloperoxidase, and NF-$\kappa$B. In addition, thymoquinone treatment in mice resulted in significant decreases in cyclooxygenase-2 and its product prostaglandin E2, as well as the immune reaction of TNF-$\alpha$. Histopathological analysis further confirmed the antiedematous and anti-inflammatory effects of thymoquinone in inflamed tissues. The authors concluded that the results support the potential use of thymoquinone to alleviate acute inflammation due to its strong antioxidant and anti-inflammatory properties in inflamed paw tissue.

### 3.2.2. *N. sativa* and Thymoquinone's Antibacterial Properties

The healing of wounds can be delayed by infections and pathological conditions such as cellular disorders, ischemia, neuropathy, and angiogenesis. This is especially true for diabetic patients. Recent research has improved our understanding of the association between wound healing and the skin microbiome [92]. The interactions between different species within the microbial environment are dynamic and can change bacterial behavior, resulting in increased virulence and delayed wound healing. A balanced or diverse microbiome, on the other hand, is necessary for effective wound healing because it inhibits pathogen development, lowering the risk of infection, persistent inflammation, and chronic wounds. Due to the complexity and dynamic nature of the skin wound-healing process,

it is necessary to use combined treatments that target both the host and the microbiome. Encouraging findings on skin microbiome modifications in wound healing and dysbiosis-related skin disorders point to novel treatment options [93]. However, further studies are needed to better understand how the skin microbiome collaborates with the host during wound healing.

In this regard, a large amount of published data suggests that the constituents of *N. sativa* seeds have the potential to modulate the immune system, which could affect the relationship between the host and parasites. In line with this, it has been reported that the active compounds of *N. sativa* oil and seeds have antimicrobial properties, including antibacterial, antifungal, anthelminthic, and antiviral effects [63]. It has been suggested that some of the antimicrobial effects of *N. sativa* seeds are due to the immunomodulatory properties of their components. *N. sativa* has been found to have antibacterial activity against several strains of bacteria, including *Escherichia coli* (*E. coli*), *Bacillus subtilis* (*B. subtilis*), *Streptococcus faecalis* (*S. faecalis*), *Staphylococcus aureus* (*S. aureus*), and *Pseudomonas aeruginosa* (*P. aeruginosa*). It also has antifungal properties and has been shown to be effective against the pathogenic yeast *Candida albicans* (*C. albicans*) and other fungi [48,94,95]. In a previous study, dithymoquinone was found to have antibacterial effects against Gram-positive bacteria [95]. Additionally, a diethyl ether extract of *N. sativa* was shown to inhibit the growth of Gram-positive bacteria *S. aureus*, as well as Gram-negative bacteria *P. aeruginosa* and *E. coli*, in a concentration-dependent manner. Notably, the extract was more effective against drug-resistant bacteria, including *Vibrio cholera* (*V. cholera*), *E. coli*, and all strains of *Shigella dysentriae* (*S. dysentriae*) [48]. The in vivo antibactericidal effectiveness of *N. sativa* seed constituents might be influenced by a variety of host conditions. When mice are injected with *C. albicans*, colonies of the fungus grow in various organs. Using this paradigm, a study of the anti-fungal impact of aqueous extract of *N. sativa* seeds revealed that treating infected mice daily for three days, beginning 24 h after *C. albicans* inoculation, significantly suppressed fungus development in the liver, spleen, and kidneys [94].

In individuals with diabetes, wound recovery can be significantly hindered by microbial infections. Khan et al. [94] tested a salve made from ostrich oil, honey, beeswax, extracts of *N. sativa*, propolis, and *Cassia angustifolia* on the wound-healing process in diabetic rats, and the results were encouraging. The results showed that the ointment had remarkable antibacterial activity against common bacteria such as *S. aureus*, *E. coli*, *Acinetobacter baumannii*, and *P. aeruginosa*. When compared to the control group, the ointment considerably accelerated wound healing and boosted collagen deposition in vivo. A histopathological evaluation also revealed the presence of hair follicles, sebaceous glands, and vessels in the group treated with the ointment. These results suggest that the ointment was successful in rapidly healing diabetic wounds and could be a suitable candidate for wound healing [94].

A recent study examined the physicochemical effects of *N. sativa* honey from the Burdur region of Turkey, as well as its antioxidant and antimicrobial properties. The honey samples showed high antimicrobial activity against several bacteria, including *S. aureus*, *E. coli*, *Chromobacterium violaceum*, *Bacillus cereus*, *Klebsiella pneumoniae*, and *Acinetobacter haemolyticus* [96].

The evidence discussed above demonstrates that the constituents of *N. sativa* seeds have anti-microbial properties against a variety of pathogens, including bacteria, viruses, helminths, and fungi. This is extremely important in practice since *N. sativa* seeds have been used historically and therapeutically in the Arab and Islamic world with no recorded harmful effects. In essence, it might be a useful co-therapeutic agent against a variety of bacteria. Additional research is needed, however, to understand the particular mechanisms of *N. sativa's* antimicrobial properties, both alone and in combination with pharmaceuticals, and to evaluate its potential therapeutic effects on other bacterial, viral, and parasitic models [97].

### 3.2.3. The Protective Impacts of *N. sativa* and Thymoquinone against Oxidative Damage

Oxidation is a chemical process that can damage cells and tissues by producing harmful molecules called reactive oxygen species (ROS). Antioxidants are substances that can stop or slow down oxidation by neutralizing or removing ROS, even when they are present in small amounts [98,99]. Antioxidants, substances that can stabilize or scavenge free radicals, can be divided into two categories: natural and synthetic. Natural antioxidants are obtained from sources such as food and plants and are widely available. These include polyphenols, carotenoids, and vitamins derived from plant materials. Synthetic antioxidants, by contrast, are artificially created and do not come from natural sources [99,100]. Fruits, vegetables, nuts, seeds, leaves, roots, and bark are abundant in antioxidants and are packed with natural compounds that aid in cell protection against free radical damage [101]. In addition to their fundamental antioxidant properties, naturally occurring antioxidants have a wide range of biological functions and provide significant nutritional benefits. They are completely safe to eat and have no known adverse effects. These antioxidants have several health advantages and can help protect cells from free radical damage.

In recent years, there has been growing interest in the use of plant-derived products for managing and treating wounds. Plant leaves, fruits, and seeds are thought to be potential antioxidant sources. Because of their quick development and abundance, leaves are one of the most important sources of antioxidants. Fruits and seeds are also well regarded for their antioxidant capacity due to their phytochemical properties and nutritional contents such as fibers, vitamins, and minerals [102].

At low concentrations, ROS can safeguard tissues from infections and promote efficient wound recovery by triggering signals for cell survival. [103]. However, when present in excess, ROS can induce cell damage and a pro-inflammatory state, leading to oxidative stress [104,105]. Polyphenols, for example, are antioxidants that can transfer electrons to other molecules, including ROS. This prevents electrons from being sequestered from other physiologically vital molecules such as proteins or DNA. Furthermore, antioxidants can initiate an intricate series of events that convert ROS into more stable compounds. As a result, these compounds contribute to the maintenance of non-toxic levels of ROS in wound tissues, which can aid in the healing process [45,106].

The anti-inflammatory, antioxidant, and antibacterial properties of *N. sativa* and its active constituent, thymoquinone [81], were found in many scientific reports to accelerate wound healing (Figure 5). For example, Bordoni et al. [107] evaluated *N. sativa* oil's antioxidant properties in inflamed adipose tissue using human preadipocytes. The total antioxidant activity measured in the supernatant of the preadipocytes showed that *N. sativa* oil has very high residual activity. Tiji et al. [108] compared the antioxidant effects of different extracts and fractions obtained from *N. sativa* seeds using hexane and acetone as solvents. They found that the extracts contained fractions with varying levels of antioxidant activity, which might be related to the presence of specific secondary metabolites in each fraction, such as polyphenols. This result is consistent with another study that reported that the methanolic extract of *N. sativa* seeds had higher antioxidant activity than the aqueous extract [109]. Ouattar et al. [110] investigated the antiradical activity of *N. sativa* extracts using different solvents and doses. They found that the crude extract had lower antiradical activity than the n-butanol or ethyl acetate extracts. These results suggest that *N. sativa* has antioxidant properties, which have been confirmed by various methods in other studies. However, the level of antioxidant activity depends on the type of extract and the fraction tested [111].

## 4. Recent Advancements in Thymoquinone's Water Solubility and Low Skin Penetration

Numerous studies have demonstrated that Thymoquinone can serve as an alternative medication for wound healing in rats, whether applied topically or systemically [112]. In a recent study, Sedik et al. evaluated the effectiveness of thymoquinone treated with cold plasma (TQcp) on wound healing in rats and compared it to thymoquinone alone [113].

A full-thickness wound model was used to assess the wound-healing potential of TQcp. The induced wound was treated twice daily with TQcp and with thymoquinone for 7 days, starting immediately after excision. The results showed that TQcp improved skin healing by increasing the production of hyaluronic acid and collagen type I, reducing the skin content of TNF-$\alpha$, and inhibiting hypertrophic scarring by up-regulating the skin content of transforming growth factor beta (TGF-$\beta$). Additionally, TQcp increased the levels of IL-10 (an anti-inflammatory cytokine), alpha smooth muscle actin, and VEGF, demonstrating its potential for wound healing. Vascular endothelial growth factor (VEGF) and TGF-$\beta$ are two key growth factors that play a role in skin repair. During inflammation, VEGF is produced by various cells, leading to endothelial migration, the synthesis of chemotactic agents, proliferation, the formation of granulation tissue, and angiogenesis [114,115]. TGF-$\beta$ is a crucial component in the wound-healing process, as it stimulates angiogenesis, fibroblast proliferation, and collagen synthesis. In the early stages of healing, the release of TGF-$\beta$ triggers the mobilization of inflammatory cells to the site of the wound [116]. TGF-$\beta$ stimulates cells to increase the synthesis of proteins from the extracellular matrix (ECM), resulting in a reduction in collagen proteases. After a skin injury, new connective tissue must be formed. During this process, fibroblasts are activated and begin to multiply, moving to the site of the wound to produce various matrix proteins, including fibronectin and collagen. Fibroblasts then differentiate into myofibroblasts and produce increased levels of alpha-smooth muscle actin ($\alpha$-SMA), which is considered a specific marker of myofibroblasts [117]. The findings of Sedik et al. [113] demonstrated that TQcp had a significantly greater potential for wound healing than thymoquinone alone.

Another recent study was conducted to improve the therapeutic efficacy of thymoquinone by designing and characterizing a nanoemulsion-based hydrogel system [112]. The aim of this nanotechnology-mediated drug delivery approach was to enhance the solubility and skin permeability of thymoquinone. The researchers used *N. sativa* oil, which contains thymoquinone, to increase the amount of the drug that could be loaded into the nanoemulgel. The results showed that the nanoemulgel system of thymoquinone improved the delivery and deposition of the drug into the skin after topical application, compared to a conventional hydrogel system. The nanoemulgel technology also promoted faster and earlier wound healing in Wistar rats than the standard thymoquinone hydrogel and displayed similar healing benefits as a commercial silver sulfadiazine cream. A histopathology revealed the formation of a thick epidermal layer, papillary dermis, and organized collagen fibers in the fixed tissues of animals treated with the nanoemulgel system [112]. These preliminary data indicate that the topical administration of thymoquinone with nanoemulgel technology is a potential alternative for accelerating wound healing.

Gomaa and colleagues [118] carried out a study to assess the efficacy of composite nanofibers loaded with thymoquinone in healing incisional wounds in mice [118]. The study found that using a thymoquinone-loaded dressing on the wounds of mice resulted in optimal wound closure. This treatment led to a thicker granulation tissue and quicker epithelial migration compared to the control group. Additionally, thymoquinone administration resulted in the maximum collagen deposition in the healed tissue. The study indicated that thymoquinone enhances wound healing by acting as an anti-inflammatory [118]. Overall, *N. sativa* and its bioactive component, thymoquinone, have been demonstrated to be profitable in accelerating the healing of incisional wounds.

## 5. The Impact of Thymoquinone on the Balance between Wound Healing and Tissue Fibrosis

The balance between wound healing and tissue fibrosis, which is dependent on the state of inflammation, describes the transition of type 2 epithelial to mesenchymal (EMT2). Natural substances include *N. sativa* and its bioactive component, thymoquinone, have been reported to affect EMT2 in various studies. Abid Nordin et al., 2019 [119] conducted a systematic review of the literature to summarize the effects of *N. sativa* and thymoquinone on EMT2 events. They searched for relevant articles in three databases and found 22 studies

that met their inclusion criteria. The review showed that most of the studies reported positive outcomes of *N. sativa* and thymoquinone treatments on wound healing, tissue inflammation, and organ fibrosis. These treatments were found to modulate the EMT2 process by influencing the expression of epithelial and mesenchymal markers, cytokines, fibrotic factors, and oxidative stress parameters. The review concluded that *N. sativa* and thymoquinone have potential therapeutic benefits for EMT2-related conditions [119].

The molecular processes involved in the transition from epithelial to mesenchymal tissue and wound healing are similar, particularly in the context of skin wounds [120]. During normal wound healing, changes in the phenotype of cutaneous tissue following EMT allow for the reepithelialisation of the epidermal layer to progress [121]. In a study on incisional wounds, it was found that the topical application of *N. sativa* cream had a positive effect on the healing of full-thickness wounds in rats. The groups treated with *N. sativa* cream had the smallest-scale wound size at the end of the experiment, with a 99.69% reduction in wound area, compared to an 81.35% reduction in the control groups. Histological analysis showed that *N. sativa* cream reduced inflammatory cell infiltration and cell edema. Additionally, the highest levels of antioxidants were observed in the tissues of rats treated with *N. sativa* cream, suggesting a correlation between the antioxidant effects of *N. sativa* and its benefits for wound healing [122].

Delayed or non-healing wounds are a common complication of diabetes mellitus. This is largely due to changes in both macrovascular and microvascular systems, which can end up with complications including poor circulation, decreased immunity, and disturbed cellular metabolism [123]. In a study of diabetic wounds, thymoquinone showed slower overall wound contraction than the control group. However, a histological investigation revealed that thymoquinone accelerated healing on day 3, before the pace of healing began to slow. The observation of lowered inflammatory cell infiltration on day 3 shows that thymoquinone accelerated wound healing during the inflammatory phase but slowed when the healing process progressed to the proliferative phase. The anti-angiogenic effects of thymoquinone are thought to be the explanation for the delayed healing during the proliferative phase [124].

In a study by Apaydin et al. [125], researchers investigated the use of *N. sativa* essential oil as a topical agent for treating diabetic wounds in rats. The results showed that STZ application significantly increased lipid peroxidation and oxidative stress in the rats. In addition, levels of GSH, GPx, SOD, and CAT in the plasma and wound tissues were decreased, while MDA levels were increased compared to the control group. Treatment with *N. sativa* essential oil increased levels of GSH, GPx, SOD, and CAT, while decreasing MDA levels compared to the diabetes group [125]. These changes in biochemical parameters were directly proportional to histopathological changes in the wound tissues. The study suggests that *N. sativa* essential oil may reduce lipid peroxidation, oxidative stress, and associated complications, and may play a beneficial role in the treatment of diabetic wounds.

Nourbar et al. [8] conducted a study to assess the effects of *N. sativa* hydroethanolic extract on wound healing in diabetic male rats. The study showed that the average healing time for untreated diabetic and phenytoin-treated diabetic groups was 27 and 24 days, respectively. Wounds in the non-diabetic untreated, sham, and phenytoin-treated non-diabetic groups were completely healed on days 23, 24, and 21. The group treated with a 40% *N. sativa* extract had the shortest healing time (15 days), followed by the group treated with a 20% *N. sativa* extract (18 days). These two groups had the smallest average wound area during the study, significantly different from the control groups' average wound area. *N. sativa* extract significantly enhanced wound healing in diabetic rats compared to control groups [8,125]. While the precise mechanism for this accelerated healing has not been investigated, it is believed that the anti-inflammatory and antibacterial properties of *N. sativa* played a role in this outcome.

In summary, these studies demonstrated that a topical administration of *N. sativa* extract may decrease inflammation and speed up wound healing in full-thickness skin wounds. However, before *N. sativa* can be claimed for therapeutic application in diabetic

patients, prospective clinical trials must be conducted to explore its safety, effectiveness, and potential mechanisms of action on wound healing (Figure 5).

## 6. Concluding Remarks and Future Perspectives

The black seed, also known as *N. sativa*, is a medicinal plant that has been gaining recognition as a miracle herb due to its rich historical background and wide spectrum of pharmacological potential. Research has revealed that the seed of *N. sativa*, its oil, and thymoquinone possess remarkable pharmacological activity, both in vitro and in vivo, in the acceleration of the four phases of wound healing. In addition, this review identifies strong scientific evidence supporting the anti-inflammatory, antioxidant, and antimicrobial properties of *N. sativa* extract and thymoquinone for the treatment of various types of cutaneous wounds [126,127] (Figure 5). This collection of literature presents the preclinical treatment of thymoquinone as an alternative medicine for wound healing. The pharmacological potential of thymoquinone has been the subject of several clinical trials investigating its therapeutic effects on hypoglycemia, hypolipidemia, and bronchodilation, and it has been approved for further drug development. However, no clinical trials have been conducted on its effects on wound healing.

There is an urgent need for additional studies, involving both animal models and clinical trials, to examine the cellular and molecular mechanisms of thymoquinone in wound healing. Future research should concentrate on improving the delivery process. Furthermore, the therapeutic benefits of *N. sativa* and thymoquinone need to be evaluated through randomized controlled trials (RCTs) to support evidence-based clinical practice. In the last few years, many meta-analyses based on the RCTs of *N. sativa* have explored its effects in clinical settings. Recent systematic meta-analyses aimed to evaluate the reporting and methodological quality and grade the available evidence of associations between *N. sativa* and health assessments. This meta-analysis included 20 eligible meta-analyses published in peer-reviewed journals from 2013 to 2021. The general quality of the methodology was notably subpar, with a single study of average quality, four of inferior quality, and 15 of extremely poor quality. Enhancements in the reporting quality are necessary for items two, five, eight, nine, 15, and 24. Among the 110 indicators for evidence quality, five were rated as moderate, 17 as low, and a significant majority of 88 as extremely low. The main factors leading to downgrading were risk of bias, inconsistency, and imprecision [128]. Therefore, more research and prospective clinical trials are needed before it can be used as a pharmaceutical therapy for wound healing. Additionally, recent computational analysis of phytochemicals' pharmacokinetic and toxicological profiles should be applied, using novel approach methodologies [129].

**Author Contributions:** Conceptualization, B.S.; resources, O.S.; data curation, O.S.; writing—B.S. and A.K.; writing—review and editing, B.S., A.K. and O.S. All authors have read and agreed to the published version of the manuscript.

**Funding:** This research received no external funding.

**Conflicts of Interest:** Author Omar Said is employed by the company Beleaf Pharma. The remaining authors declare that the research was conducted in the absence of any commercial or financial relationships that could be construed as a potential conflict of interest.

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
