# Peer review of "How Thymoquinone from Nigella sativa Accelerates Wound Healing through Multiple Mechanisms and Targets"

_cimb, doi:10.3390/cimb45110567_

Round 1
Reviewer 1 Report
Comments and Suggestions for Authors
The manuscript generally offers a nice overview of the field.
However, several improvements are needed:
Italic words should be used when mentioning species (abstract and title mostly). Same for "in vitro" and "in vivo" terms.
Figure 4 - basic chemical structure - not needed
figure 5 - too basic, not needed, could be listed in the text
ref. 75 and 76 (page 10) have different font and 76 is bolded (?)
page 10 - why is "in animal studies" italic?
Figure 6 and 7 should be revised and constructed in a standard scientific way.
Figures should be explained well in the text, not presented in a basic way as too large pictures
Comments on the Quality of English LanguageEnglish language is fine/minor corrections are needed.
Author Response
Thank you for providing me with the opportunity to revise my manuscript. I have carefully considered and addressed all comments and suggestions in my revision. I am open to further revisions if you have any additional requests or suggestions. Below is my point-by-point response to the reviewers’ comments:
Reviewer 1
Italic words should be used when mentioning species (abstract and title mostly). Same for "in vitro" and "in vivo" terms.
Done
Figure 4 - basic chemical structure - not needed
Figure deleted
figure 5 - too basic, not needed, could be listed in the text
We agree with the reviewer that this figure is too basic. However, we believe this simplicity enhances the reader’s comprehension of the text. Nevertheless, we are open to removing it if the reviewer strongly recommends doing so.
ref. 75 and 76 (page 10) have different font and 76 is bolded (?)
Has been added Corrected
page 10 - why is "in animal studies" italic?
Has been added Corrected
Figure 6 and 7 should be revised and constructed in a standard scientific way.
Unfortunately, I do not understand what the respected reviewer means. We have published several manuscripts in the prestigious MDPI that contain similar figures. For example, Saad B. A Review of the Anti-Obesity Effects of Wild Edible Plants in the Mediterranean Diet and Their Active Compounds: From Traditional Uses to Action Mechanisms and Therapeutic Targets. International Journal of Molecular Sciences. 2023 Aug 10;24(16):12641.
Figures should be explained well in the text, not presented in a basic way as too large pictures
Requested data has been added in Figure legends
Reviewer 2 Report
Comments and Suggestions for Authors
This review paper explores the potential of thymoquinone, a compound derived from Nigella sativa, for accelerating wound healing through its antimicrobial, anti-inflammatory, and antineoplastic properties. While it finds sufficient evidence supporting thymoquinone's effectiveness, it emphasizes the need for further research to determine optimal concentrations, administration routes, potential side effects, and its performance in clinical trials. Approach is novel and research is presented well, however, there are many major issues, which are unclear in current draft. Addressing these comments in the paper will enhance its clarity, credibility, and relevance to the scientific community. To improve the paper my comments are appended below
1. Introduction Clarity: The introduction provides a good overview of the topic but could be more concise. It should clearly state the objectives and the importance of the research. Additionally, you could briefly introduce the concept of "Thymoquinone" and its potential for wound healing earlier in the introduction to captivate the reader's interest. Additionally, cite https://doi.org/10.1016/j.crtox.2023.100118 a recent report with the sentence ending with ´…medicinal plants used in the treatment skin diseases; traditional uses and active compounds of N. sativa.´ on page 3 line 113-114.
2. Review Scope: The study mentions an extensive review of literature and databases. To enhance the quality of the review, please specify the time range and the number of relevant papers reviewed to ensure that it represents the most up-to-date information.
3. Citation and Source Evaluation: Assess the quality of the sources used in the review. Mention the criteria used to select and evaluate the papers. This would help in justifying the reliability and relevance of the information gathered.
4. Quantitative Data: The paper lacks specific data such as the number of studies reviewed, the sample sizes in clinical trials, and the range of concentrations studied. Including such quantitative data in the abstract can provide a clearer picture of the evidence.
5. Mechanisms Clarification: The paper mentions the "mechanisms behind its effectiveness." To enhance clarity, elaborate on the specific cellular and molecular mechanisms. You may include a brief overview of what these mechanisms are and how thymoquinone affects them.
6. Potential Side Effects: It is commendable to mention the need for investigating potential side effects. To add depth to the review, briefly discuss known side effects or potential adverse reactions associated with thymoquinone, even if they are limited. Also, cite https://doi.org/10.3390/livers3030032 on page 7 along with reference 61-62 with the sentence ´…..and carotene, which the liver converts into vitamin A´ to update the references list.
7. Clinical Trials Description: Provide a brief summary or examples of clinical trials that have explored the use of thymoquinone in wound healing. This would give readers an idea of the current state of clinical research in this field.
8. Concentration and Administration Optimization: Suggest discussing the various concentrations of thymoquinone tested in different studies and their outcomes. Additionally, emphasize the importance of optimizing the route of administration for better therapeutic efficacy.
9. Consensus and Controversies: The statement, "there is no consensus on the most effective concentrations," could be expanded to include a brief discussion of the reasons behind the lack of consensus and any controversies in the field.
10. Evaluation of Study Designs: Provide some insight into the types of studies you reviewed. Are they mostly in vitro, in vivo, or clinical studies? Assess the strengths and weaknesses of these study designs and discuss their implications for the conclusions drawn.
11. Recommendation for Future Research: In the conclusion of the paper, explicitly outline specific areas or questions for future research, such as exploring novel delivery methods or identifying potential synergies with other wound healing agents.
12. Language and Clarity: Lastly, ensure the review is written in clear, concise, and unambiguous language. Avoid overly complex sentences or jargon that may impede comprehension for readers who are not specialists in the field.
Comments on the Quality of English Language
Moderate editing of English language required
Author Response
Thank you for providing me with the opportunity to revise my manuscript and for the valuable comments from the reviewer. I have carefully considered and addressed all comments and suggestions in my revision. I am open to further revisions if you have any additional requests or suggestions. Below is my point-by-point response to the reviewers’ comments:
- Introduction Clarity: The introduction provides a good overview of the topic but could be more concise. It should clearly state the objectives and the importance of the research. Additionally, you could briefly introduce the concept of "Thymoquinone" and its potential for wound healing earlier in the introduction to captivate the reader's interest. Additionally, cite https://doi.org/10.1016/j.crtox.2023.100118 a recent report with the sentence ending with ´…medicinal plants used in the treatment skin diseases; traditional uses and active compounds of N. sativa.´ on page 3 line 113-114.
The concept of thymoquinone and its potential for wound healing has been added in the Introduction
Rai M, Singh AV, Paudel N, Kanase A, Falletta E, Kerkar P, Heyda J, Barghash RF, Singh SP, Soos M. Herbal concoction Unveiled: A computational analysis of phytochemicals' pharmacokinetic and toxicological profiles using novel approach methodologies (NAMs). Current Research in Toxicology. 2023 Jan 1;5:100118. Has been cited in the concluding remarks
- Review Scope: The study mentions an extensive review of literature and databases. To enhance the quality of the review, please specify the time range and the number of relevant papers reviewed to ensure that it represents the most up-to-date information.
The requested data has been added in the Abstract and at the end of the Introduction
- Citation and Source Evaluation: Assess the quality of the sources used in the review. Mention the criteria used to select and evaluate the papers. This would help in justifying the reliability and relevance of the information gathered.
The requested data has been added in the Abstract and at the end of the Introduction
- 4. Quantitative Data: The paper lacks specific data such as the number of studies reviewed, the sample sizes in clinical trials, and the range of concentrations studied. Including such quantitative data in the abstract can provide a clearer picture of the evidence.
The requested data has been added in the Abstract and at the end of the Introduction
- Mechanisms Clarification: The paper mentions the "mechanisms behind its effectiveness." To enhance clarity, elaborate on the specific cellular and molecular mechanisms. You may include a brief overview of what these mechanisms are and how thymoquinone affects them.
The requested data has been added to section ‘3.2. Wound Healing Beneficial Effects of Thymoquinone’ and is discussed in detail in the following sections. In addition, Figure 5 summarizes these mechanisms
- Potential Side Effects: It is commendable to mention the need for investigating potential side effects. To add depth to the review, briefly discuss known side effects or potential adverse reactions associated with thymoquinone, even if they are limited. Also, cite https://doi.org/10.3390/livers3030032 on page 7 along with reference 61-62 with the sentence ´…..and carotene, which the liver converts into vitamin A´ to update the references list.
The requested data has been added at page 8
- Clinical Trials Description: Provide a brief summary or examples of clinical trials that have explored the use of thymoquinone in wound healing. This would give readers an idea of the current state of clinical research in this field.
The requested data has been added in the Concluding remarks and future perspectives
- Concentration and Administration Optimization: Suggest discussing the various concentrations of thymoquinone tested in different studies and their outcomes. Additionally, emphasize the importance of optimizing the route of administration for better therapeutic efficacy.
The requested data has been added in 3.1 Traditional uses and active compounds of N. sativa
- Consensus and Controversies: The statement, "there is no consensus on the most effective concentrations," could be expanded to include a brief discussion of the reasons behind the lack of consensus and any controversies in the field.
The requested data has been added in the Concluding remarks and future perspectives
- Evaluation of Study Designs: Provide some insight into the types of studies you reviewed. Are they mostly in vitro, in vivo, or clinical studies? Assess the strengths and weaknesses of these study designs and discuss their implications for the conclusions drawn.
The requested data has been added at page 4
- Recommendation for Future Research: In the conclusion of the paper, explicitly outline specific areas or questions for future research, such as exploring novel delivery methods or identifying potential synergies with other wound healing agents.
The requested data has been added in the Concluding remarks and future perspectives
- Language and Clarity: Lastly, ensure the review is written in clear, concise, and unambiguous language. Avoid overly complex sentences or jargon that may impede comprehension for readers who are not specialists in the field.
Done
Round 2
Reviewer 1 Report
Comments and Suggestions for Authors
The authors made most of the changes as suggested by reviewers. However, issues with figures are still present. The new Figure no. 5 (with regards to responses): does scientifically correct mean the following: "increased receptor TRL-4" or "increased enzymes" or similar? In my best goodwill, I would suggest omitting the Figure and presenting the changes in the text, using standard scientific language, for example, " increased expression of..." or "increased enzymatic activity/concentration of...".
Comments on the Quality of English LanguageEnglish needs some polishing.
Author Response
Many thanks for your suggestion. We have removed Figure 5 and added the data to the text

Reviewer 2 Report
Comments and Suggestions for Authors
accept
Author Response
Many thanks for your recommendation to accept the manuscript
Round 3
Reviewer 1 Report
Comments and Suggestions for Authors
The manuscript has been improved.
Comments on the Quality of English LanguageMinor/moderate changes needed.
Author Response
Dear Reviewer,
We greatly appreciate your suggestion. Rest assured, we have thoroughly reviewed our manuscript for English language accuracy. Thank you for your attention to detail.
We have revised Figure 5